# Galectin-9 and Interferon-Gamma Are Released by Natural Killer Cells upon Activation with Interferon-Alpha and Orchestrate the Suppression of Hepatitis C Virus Infection

**DOI:** 10.3390/v14071538

**Published:** 2022-07-14

**Authors:** Anna Paola Carreca, Massimiliano Gaetani, Rosalia Busà, Maria Giovanna Francipane, Maria Rita Gulotta, Ugo Perricone, Gioacchin Iannolo, Giovanna Russelli, Claudia Carcione, Pier Giulio Conaldi, Ester Badami

**Affiliations:** 1Fondazione Ri.MED, 90133 Palermo, Italy; apcarreca@fondazionerimed.com (A.P.C.); massimiliano.gaetani@ki.se (M.G.); mgfrancipane@fondazionerimed.com (M.G.F.); mrgulotta@fondazionerimed.com (M.R.G.); uperricone@fondazionerimed.com (U.P.); ccarcione@fondazionerimed.com (C.C.); 2Department of Research, IRCCS-ISMETT, Istituto Mediterraneo per i Trapianti e Terapie Ad Alta Specializzazione, 90127 Palermo, Italy; rbusa@ismett.edu (R.B.); giannolo@ismett.edu (G.I.); grusselli@ismett.edu (G.R.); pgconaldi@ismett.edu (P.G.C.); 3Chemical Proteomics, Department of Medical Biochemistry and Biophysics, Karolinska Institute, Biomedicum, SE-17177 Stockholm, Sweden; 4SciLifeLab (Science for Life Laboratory), SE-17177 Stockholm, Sweden

**Keywords:** HCV, NK cells, IFN-α, galectin-9, IFN-γ

## Abstract

Natural killer (NK) cells mount an immune response against hepatitis C virus (HCV) infection and can be activated by several cytokines, including interleukin-2 (IL-2), IL-15, and interferon-alpha (IFN-α). By exploiting the Huh7.5 hepatoma cell line infected with the HCV JFH1 genome, we provide novel insights into the antiviral effector functions of human primary NK cells after cytokine stimulation. NK cells activated with IFN-α (IFNα-NKs) had enhanced contact-dependent and -independent responses as compared with NK cells activated with IL-2/IL-15 (IL2/IL15-NKs) and could inhibit HCV replication both in vitro and in vivo. Importantly, IFN-α, but not IL-2/IL-15, protected NK cells from the functional inhibition exerted by HCV. By performing flow cytometry, multiplex cytokine profiling, and mass-spectrometry-based proteomics, we discovered that IFNα-NKs secreted high levels of galectin-9 and interferon-gamma (IFN-γ), and by conducting neutralization assays, we confirmed the major role of these molecules in HCV suppression. We speculated that galectin-9 might act extracellularly to inhibit HCV binding to host cells and downstream infection. In silico approaches predicted the binding of HCV envelope protein E2 to galectin-9 carbohydrate-recognition domains, and co-immunoprecipitation assays confirmed physical interaction. IFN-γ, on the other hand, triggered the intracellular expressions of two antiviral gate-keepers in target cells, namely, myxovirus-1 (MX1) and interferon-induced protein with tetratricopeptide repeats 1 (IFIT1). Collectively, our data add more complexity to the antiviral innate response mediated by NK cells and highlight galectin-9 as a key molecule that might be exploited to neutralize productive viral infection.

## 1. Introduction

Hepatitis C virus (HCV) is an RNA virus that specifically infects hepatocytes. In 70–80% of cases, acute HCV infection degenerates into chronic inflammation, which is accompanied by progressive hepatic fibrosis and increased cancer risk [1]. Traditionally, chronic hepatitis C (CHC) patients have been treated with interferon-alpha (IFN-α) or its PEGylated form (PEG-IFN-α) alone or in combination with nucleoside analogue ribavirin (RBV), with sustained virological responses in 50–60% of cases [2]. The relatively poor responsiveness to standard (PEG-) IFN-α/RBV regimens, combined with the high number of adverse effects, has prompted efforts to develop new treatments. Direct-acting antivirals (DAAs) have emerged as alternative antiviral drugs [3]. After the first-generation DAAs, which were added to therapeutic schemes, combinations of second-generation DAAs more recently proved to be highly effective and well tolerated, opening a new era of IFN-free HCV treatments. DAA-based therapy is, however, not free of challenges. Among these are high costs, high pill burdens, stringent dosing requirements, and the emergence of resistant viral mutants [4].

Potentiating or restoring natural antiviral immunity could be a promising strategy to control HCV spread alongside standard treatment regimens. Successful antiviral immunity requires a coordinated response from both innate and adaptive systems, and natural killer (NK) cells, which are at the interface of these systems, have a well-established role in HCV response. Both liver-infiltrating and resident NK cells (these latter accounting for up to 50% of lymphocytes in the liver [5]) contribute to HCV suppression. NK-cell activity is driven by the expressions of activating and inhibitory receptors, which are under the control of several cytokines, including interleukin-21 (IL-21), IL-23, IL-27, IL18, IL-12, IL-15, IL2, and IFN-α [6,7]. In NK cells, IFN-α triggers the expression of membrane-bound tumor necrosis factor-related apoptosis-inducing ligand (TRAIL) [8], and degranulation and cytokine release through the activation of the signal transducer and activator of transcription 1 (STAT1) signaling pathway [9].

During HCV acute infection, NK cells are activated, exhibit direct cytotoxic functions, and secrete interferon-gamma (IFN-γ) [10], thus contributing to the containment of viral infection [11]. However, during chronic infection, these functions are impaired [12], despite the fact that NK cells still retain an activated phenotype [13]. This impairment likely results in increased viral replication and liver damage [14]. Interestingly, in vitro studies have suggested that HCV proteins including the core protein inhibit NK-cell maturation and functions [15,16].

An improved understanding of the cellular and molecular mechanisms of cytokine-enhanced NK-cell response to infection and cancer would lead to a resurgence of the interest in the clinical use of cytokines that sustain and/or activate NK-cell potential. Moreover, dissecting impaired NK-cell function may help develop novel immunotherapeutic strategies. Using an HCV cell culture model [17], we examined whether NK-cell effector functions could be enhanced in response to specific cytokines such as IFN-α, historically used as an antiviral therapy [18]. NK cells activated in vitro with IFN-α (IFNα-NKs) could stably clear HCV-infected cells, and block viral replication and de novo infection. In contrast, treatment with IFN-α alone resulted in a weaker and short-term viral suppression. Similarly, NK cells activated with IL-2/IL-15 (IL2/IL15-NKs) showed lower cytotoxicity against HCV-infected cells as compared with IFNα-NKs. Importantly, IFNα-NKs, but not IL2/IL15-NKs, were resistant to the functional impairment induced by HCV. The increased antiviral functions of IFNα-NKs were also observed in vivo using the humanized chimeric urokinase-type plasminogen activator/severe combined immunodeficiency (uPA/SCID) mouse model, which supports the whole HCV biological cycle [19].

Through the analysis of the NK-cell-secreted factors using both a multiplex cytokine assay on conditioned media and mass-spectrometry-based proteomics on secretome [20,21], we identified two main biomarkers: IFN-γ and β-galactose-binding lectin galectin-9. In our system, galectin-9 suppressed HCV infection independently from its receptor T cell immunoglobulin and mucin protein 3 (TIM-3) [22]. We hypothesized that galectin-9 might bind to HCV virions through its carbohydrate-binding region, thus preventing viral entry into host cells and infection. We used in silico approaches to prove our hypothesis, predicting a binding of HCV envelope protein E2 to galectin-9 carbohydrate-recognition domains, which was finally confirmed with a co-immunoprecipitation assay. Importantly, galectin-9 and IFN-γ together suppressed HCV infection. Furthermore, using mass-spectrometry-based proteomics, we found that IFNα-NKs triggered the expression of antiviral gatekeepers myxovirus-1 (MX1) and interferon-induced protein with tetratricopeptide repeats 1 (IFIT1) in HCV-infected cells, through a mechanism mediated by IFN-γ.

Our data add more complexity to the NK-cell response to HCV infection, highlighting the crucial role of galectin-9 in the suppression of the HCV biological cycle.

## 2. Materials and Methods

### 2.1. Ethics Statement

In vitro experiments using patient-derived cells were conducted in accordance with the principles outlined in the Declaration of Helsinki and with the approval of ISMETT Institutional Research Review Board (IRRB approval No. 14/15; 7 April 2015). The animal protocol was approved by Ethics Committee for Animal Research of Faculty of Medicine at Ghent University [19].

### 2.2. Cell Culture and Treatments

Leucocytes were isolated from the perfusion fluid of liver explants, as previously described [23,24]. Primary CD3-CD56+ NK cells were isolated from buffy coats using a human NK-cell isolation kit (Miltenyi Biotec, Bergisch Gladbach, Germany). Cell isolates with <90% viability, as assessed with the trypan blue exclusion test, and <95% purity were excluded from the study. NK cells were cultured immediately after isolation in MACS NK medium (Miltenyi Biotec, Bergisch Gladbach, Germany) supplemented with 500 IU/mL IL-2 (Proleukin, Chiron, Emeryville, CA, USA) and 20 ng/mL IL-15 (Miltenyi Biotec, Bergisch Gladbach, Germany), in the presence of NK activation/expansion beads for five days (Miltenyi Biotec, Bergisch Gladbach, Germany), or alternatively, they were cryopreserved. The concentrations of IL-2 and IL-15 were chosen based on titration assays (IL-2, 250–1000 IU/mL; IL-15, 10–30 ng/mL).

To investigate NK-cell antiviral effector functions, cells were further stimulated overnight with IL-2/IL-15 or 1 µg/mL IFN-α (Origene, Herford, Germany). Before downstream applications, NK cells were thoroughly washed.

Where indicated, NK cells were exposed to HCV virions for four hours, or treated with 10 µg/mL blocking anti-TRAIL (CD253) antibody (clone RIK-2; BD Biosciences, Europe) [25], 10 µg/mL anti-galectin-9 antibody (clone 9M1-3) [26], or 10 µg/mL anti-human IFN-γ (clone B27) [27] (Biolegend, San Diego, CA, USA). Huh7.5 cells were transfected with HCV JFH1 (genotype 2a), which, for some experiments, was tagged with the Venus reporter, an enhanced version of Yellow Fluorescent Protein (YFP) [28]. Transfections were performed using electroporation (Nucleofector 4D; Lonza, Euroclone, Pero, Italy), as previously described [24]. Control or HCV-infected Huh7.5 cells were co-cultured with NK cells in direct contact at effector:target (E:T) ratios of 20:1, 10:1, 5:1, 2.5:1, or 1:1, or alternatively, they were cultured using Millicell cell culture inserts (Millipore, Milan, Italy) at an E:T ratio of 10:1.

The pFK-Luc-Jc1 and pFK-Venus-Jc1 plasmids (genotype 2a) and Huh7.5 cells were a kind gift from Prof. R. Bartenschlager with the authorization of Apath L.L.C., NY.

### 2.3. Cell Cytotoxicity and Proliferation Assays

A four-hour chromium-51 (^51^Cr) release assay (Perkin Elmer Italia, Milano, Italy) was used to measure the specific lysis of target cells exposed to decreasing ratios of NK cells (E:T ratios of 20:1, 10:1, 5:1, 2.5:1, or 1:1). The percent of target cell lysis was calculated as follows: (mean experimental counts per minute (cpm) − mean spontaneous cpm)/(mean maximum cpm − mean spontaneous cpm) ×  100%.

Cell proliferation was assessed using an MTS colorimetric assay following the manufacturer’s instructions (Abcam, Cambridge, UK).

### 2.4. Flow Cytometry Staining

Aliquots of 1 × 10^6^ NK cells were analyzed for surface markers using a panel of prediluted fluorochrome-conjugated anti-human monoclonal antibodies. FITC-CD3, PE-Cy™7-IFNγ, FITC-CD107a, PE-NKp44, and PerCP-Cy™5.5-NKG2D were purchased from BD Biosciences. PerCP-CD3, PE-CD253/TRAIL, Alexa Fluor-700-TNFα, APC-CD56, APC-NKp30, PE-Cy7-NKp46, and Alexa Fluor-700-TIM-3 were purchased from eBioscience (San Diego, CA, USA). PE-Galectin-9 was purchased from Miltenyi Biotec (Bergisch Gladbach, Germany). Samples were run on a FACSAria II dual-laser 8-color cytometer (BD Biosciences, Europe), and data were analyzed with FACSDiva software 6.1.3.

### 2.5. HCV Core Protein (HCV-Ag) Quantification and Staining

The presence of HCV-Ag in the supernatant of HCV-infected Huh7.5 cells was assessed using two-step chemiluminescent microparticle immunoassay ARCHITECT HCV Ag on ARCHITECT-i2000R Immunoassay Analyzer (Abbott Diagnostics, Lake Forest, IL, USA). For HCV-Ag immunostaining, a mouse anti-HCV core antibody (Abcam, Cambridge, UK) was used [29]. Images were acquired with a confocal microscope (TCS SP5 II; Leica, Wetzlar, Germany) and analyzed using ImageJ software 1.53 K (Rasband, W.S., ImageJ, U.S. National Institutes of Health, Bethesda, MA, USA; https://imagej.nih.gov/ij/, 1997–2018 (accessed on 9 September 2021)).

### 2.6. ELISA and Multiplex Analysis of Cytokines

Cytokines in NK-cell-conditioned media were quantified using Bead CD8+ T cells Magnetic Bead Panel, Cytokine/Chemokine Magnetic Bead Panel, Cytokine/Chemokine Magnetic Panel III, Th17 Magnetic Bead Panel, or Cytokine/Chemokine Panel II (all human; Merck Millipore, Burlington, MA, USA). Plates were read on Luminex^®^ 200 system, and data were analyzed with Procarta Plex™ Analyst 1.0 (Affymetrix, Waltham, MA, USA). Cytokine levels were normalized to the number of plated cells. The levels of human galectin-9 in the conditioned media of NK-cells were quantified with a sandwich ELISA following the manufacturer’s instructions (RayBiotech, Peachtree Corners, GA, USA). Plates were read on a Spark microwell plate reader (Tecan, Mannedorf, Switzerland).

### 2.7. In Vivo Study

For the in vivo study, 2–3 week old SCID-beige/Alb-uPA mice (*n* = 6) were transplanted with 1 × 10^6^ human primary hepatocytes (donor code HH223; BD Biosciences, Europe). Twelve weeks later, mice were infected intraperitoneally with 2 × 10^4^ mH77c HCV-JFH1 virions [19]. The following day, mice received 2 × 10^7^ NK cells intravenously (NK cells were previously activated or not with IL-2/IL-15 or IFN-α) (*n* = 2/group). A second injection with 10^7^ NK cells activated as before followed one week later [30]. At regular intervals, mice were bled and plasma was stored at −70 °C until further analysis. Six weeks after the first NK-cell injection, all mice were sacrificed, and their livers were collected and processed for RNA and histological analyses.

### 2.8. Quantification of Human Albumin and HCV Viremia

Human albumin in mouse plasma was measured using an in-house sandwich ELISA as described elsewhere [31]. HCV RNA levels were quantified using a Cobas Amplicor HCV Monitor test v2.0 (Roche Diagnostics, Monza, Italy).

### 2.9. Immunohistochemical Staining

Immunohistochemistry for HCV NS3 (VWR, Radnor, PA, USA) or CD56 (Abcam, Cambridge, UK) was performed on paraffin sections of livers from all mouse groups using antigen retrieval with citrate buffer and a VECTASTAIN^®^ Elite ABC peroxidase kit, in combination with DAB substrate (DAKO, Copenhagen, Denmark) and Mayer’s hematoxylin.

### 2.10. Strand-Specific RT-PCR for HCV Detection

For the detection of the negative-strand HCV RNA, total RNA obtained from HCV-infected Huh7.5 cells or mouse livers was retro-transcribed using Reverse Transcription System (Promega, Madison, WI, USA) and amplified using 1.5 μM HCV1 sense primer, as previously described [32].

### 2.11. Secretome Preparation and Quantitative Proteomics

Secretome was prepared by concentrating conditioned media of NK or Huh7.5 cells by precipitation with 15% trichloroacetic acid (Sigma Aldrich, St. Louis, MO, USA) on ice for 15 min, followed by centrifugation at 14,000× *g* for 20 min at 4 °C. Samples were processed and analyzed with tandem mass tag (TMT)-based quantitative proteomics and combined into a multiplex sample. The TMT multiplexed sample was split in two fractions (F1 and F2) and run on a Q-Exactive mass spectrometer (MS) equipped with a Nanospray Flex ion source and coupled with an UHPLC Ultimate 3000 system (ThermoFisher Scientific, Waltham, MA, USA). Raw data of both sets of fractions F1 and F2 were analyzed using Proteome Discoverer 2.1 software (ThermoFisher Scientific, Waltham, MA, USA). The MS/MS peptide spectrum match was performed using both the Sequest HT and Mascot database search engines in series against the SwissProt human proteome database (taxonomy identification number 9606), using thresholds of two trypsin mis-cleavages and a false discovery rate of 0.05.

### 2.12. Protein Structure Preparation Using Protein Data Bank (PDB)

The structures of the galectin-9 carbohydrate recognition domains (CRDs) (PDB IDs: 2ZHM and 2ZHN) [33] and HCV E2 (PDB IDs: 6MEK and 6MEJ) [34] were downloaded from PDB [35] and optimized using the “Protein preparation” tool of the Schrödinger suite (Schrödinger Inc., New York, NY, USA; software release v2021-3) [36]. The bond orders for untemplated residues were assigned using known HET groups based on their SMILES (Simplified Molecular Input Line Entry System) strings in the Chemical Component Dictionary. Hydrogens were added to the structures, zero-order bonds between metals and nearby atoms were also added, and formal charges to metals and neighboring atoms were corrected. Disulfide bonds were created according to possible geometries, and water molecules beyond 5.0 Å from any of the HET groups, including ions, were deleted. Then, the protonation and metal charge states for the ligands, cofactors, and metals were generated [37,38]. PROPKA [38] was run at pH 7.0 to optimize hydroxyl groups and aspartyl (Asn), glutamyl (Gln), and histidine (His) states.

### 2.13. Protein–Protein Docking of Galectin-9 CRDs and HCV E2

In order to perform protein–protein docking, the sugar moieties included in the PDB structures of the galectin-9 CRDs (2ZHM and 2ZHN) were deleted, and only chain A was maintained. For HCV E2, only chain C, linked to oligosaccharides, was maintained from PDB structures 6MEK and 6MEJ. These chains were used to perform two protein–protein docking screenings in parallel, one involving structures from 2ZHM and 6MEK, and another one including 2ZHN and 6MEJ. For this purpose, the tool “Protein-protein docking” of the Schrödinger suite was used. Chain A from PDB 2ZHM and 2ZHN was set as receptor, while chain C from PDB 6MEK and 6MEJ as ligand. The number of ligand rotations to attempt for docking structures was set at 70,000. Each rotation was used to find the best docking score with the receptor. The 1000 top-scoring rotations were then clustered based on the binding site by simply calculating the root-mean-square deviation (RMSD), and each cluster generated one docking structure. The maximum number of docking structures to return was set to 30. No attraction, repulsion, or distance restraints were set. Finally, the output poses were refined.

### 2.14. Molecular Dynamics (MD) Simulations of Galectin-9 CRDs–HCV E2 Complexes

We ran two MD simulations using Desmond [39]: one MD simulation of 250 ns of the 2ZHM–6MEK complex, and another MD simulation of 250 ns of the 2ZHN–6MEJ complex. Both the trajectories were generated by applying the same MD settings, which are described below. The systems were created using TIP3P [40] as a solvent model, and the orthorhombic shape box was chosen. The box side distances were set at 10 Å. Force field OPLS4 [41] was applied, and the systems were neutralized by adding Na_+_ ions. Na^+^Cl^−^ salt was added to the system at a concentration of 0.15 M. The outputs were further processed by performing MD simulations lasting 250 ns each.

Ensemble class NPT was chosen to maintain the number of atoms, the pressure, and the temperature constant for the entire trajectories. The thermostat method employed was the Nosé–Hoover chain with a relaxation time of 1.0 ps and a temperature of 300 K. The barostat method applied was Martyna–Tobias–Klein, with a relaxation time of 2.0 ps and an isotropic coupling style. The timestep for numerical integration was 2.0 fs for bonded interactions, 2.0 fs for nonbonded–near interactions (van der Waals and short-range electro-static interactions), and 6.0 fs for nonbonded–far interactions (long-range electrostatic interactions). For Coulombic interactions, a cut-off radius of 9.0 Å was tuned as a short-range method. Pressure and temperature were set at 1.01325 bar and 300 K, respectively. Finally, the systems were relaxed before beginning the simulations according to the following steps: (1) minimization with the solute restrained; (2) minimization without restraints; (3) 12 ps in the NVT ensemble with a Berendsen thermostat, temperature of 10 K, a fast temperature relaxation constant, velocity resampling every 1 ps, and nonhydrogen solute atoms restrained; (4) 12 ps in the NPT ensemble in Berendsen thermostat and barostat, temperature equal to 10 K and a pressure of 1 atm, a fast temperature relaxation constant, a slow pressure relaxation constant, velocity resampling every 1 ps, and nonhydrogen solute atoms restrained; (5) 24 ps in the NPT ensemble with Berendsen thermostat and barostat, temperature of 300 K and a pressure of 1 atm, a fast temperature relaxation constant, a slow pressure relaxation constant, velocity resampling every 1 ps, and nonhydrogen solute atoms restrained; (6) 24 ps of relaxation in NPT ensemble using Berendsen thermostat and barostat, a temperature of 300 K and a pressure of 1 atm, a fast temperature relaxation constant, and a normal pressure relaxation constant.

### 2.15. MD Frame Clustering

In order to retrieve specific contacts between amino acids of protein partners during the entire MD simulations, the frames were clustered to identify the most representative centroids to be analyzed. The RMSD matrix calculation was set using the protein backbone as a reference, the frequency of frame analysis was set to 10, and the hierarchical cluster linkage method was employed as an average. Finally, for both MD trajectories, ten clusters were generated, and the analysis is reported in the Results section.

### 2.16. Co-Immunoprecipitation Assay

Recombinant JFH1-HCV was incubated with or without 20 ng of galectin-9 (RayBiotech, Peachtree Corners, GA, USA) in a buffer containing 2.5% BSA, 1% Triton X-l00, 50 mM HEPES (pH 7.5), 150 mM NaCl, 10% glycerol, 1.5 mM MgCl2, and 5 mM EGTA in the presence of 3 μL of anti-galectin-9 (PA5-29823; Invitrogen, Waltham, MA, USA) and 30 μL of protein A-Sepharose beads (Sigma-Aldrich, St. Louis, MO, USA) for two hours with gentle rotation at +4°C. The unbound proteins were washed six times with lysis buffer [42]; then, the slurry beads were resuspended with 300 μL of PBS and evaluated with two-step chemiluminescent microparticle immunoassay ARCHITECT HCV Ag on ARCHITECT-i2000R Immunoassay Analyzer (Abbott Diagnostics, Lake Forest, IL, USA).

### 2.17. HCV Neutralization Assay

Recombinant HCV-JFH1 was pre-incubated with 0–50 ng/mL of purified human galectin-9 (RayBiotec, Peachtree Corners, GA, USA) for two hours at 37 °C in serum-free DMEM. Huh7.5 cells were seeded at 30,000 cells/well in 24-well plates and infected for six hours with the HCV-JFH1/galectin-9 mix. Media was then replaced with DMEM supplemented with 10% FBS. Infection was monitored five days later with immunofluorescence.

### 2.18. Western Blotting

Cells were collected with STET lysis buffer containing protease inhibitor cocktail (Sigma) and isolated proteins were separated with SDS-PAGE electrophoresis and later transferred on PVDF membranes (Biorad, Hercules, CA, USA). Membranes were incubated with the following antibodies: anti-galectin-9 (Invitrogen, Waltham, MA, USA), anti-MX1 antibody (Abcam, Cambridge, UK), or anti-beta actin (Santa Cruz biotechnology, Dallas, TX, USA).

### 2.19. Statistical Analysis

The statistical analyses considered both technical and biological replicates. Specific tests are described in the legends and were performed using GraphPad Prism software, v9. Statistical tests were considered significant when the *p*-value was less than 0.05 (* *p* < 0.05, ** *p* < 0.01, *** *p* < 0.001, and **** *p* < 0.0001).

## 3. Results

### 3.1. NK Cells Activated with IFN-α Had Enhanced Antiviral Functions

With the aim of studying how different cytokines regulate NK-cell antiviral functions, we first isolated CD3^-^CD56^+^ NK cells with high purity (>95%) from the cell fraction of liver perfusates from heart-beating brain-dead donors, as previously described (Appendix A) [23,24]. Freshly isolated or thawed NK cells were expanded for five days and activated overnight with IL-2/IL-15 or IFN-α.

To investigate the antiviral response of both IL2/IL15-NKs and IFNα-NKs, cells were challenged with human hepatoma HuH-7 cell line derivative Huh7.5 infected with the HCV-JFH1 replicon (from now on indicated as HCV-Huh7.5) [17]. We used the chromium 51 (^51^Cr) release assay to detect and quantitate the specific lysis of HCV-Huh7.5 cells induced by NK cells. We observed that IFNα-NKs had greater cytotoxicity against HCV-Huh7.5 cells than IL2/IL15-NKs (Figure 1A). Non-infected Huh7.5 cells, which trigger an NK-cell-mediated response due to their cancerous phenotype, served as controls (Figure 1B). Interestingly, IL2/IL15-NKs showed significantly lower cytotoxicity against HCV-Huh7.5 cells than uninfected controls (Figure 1A,B), suggesting that the presence of HCV impairs NK-cell functions. Conversely, the cytotoxic response of IFNα-NKs was of similar magnitude against both uninfected and HCV-infected Huh7.5 cells (Figure 1A,B), which suggests that IFNα-NKs might resist HCV-induced functional inhibition.

Next, we studied contact-independent NK-cell antiviral response following IL-2/IL-15 or IFN-α activation. To this purpose, a co-culture system was used, with HCV-Huh7.5 cells seeded on the bottom side of a transwell insert and IL2/IL15-NKs or IFNα-NKs plated on the upper chamber. As a readout of productive viral infection, we quantified the levels of the HCV core antigen (HCV-Ag) released in the culture medium seven days after the start of co-culture. In the presence of IFNα-NKs, the levels of HCV-Ag were barely detectable, whereas they were variably reduced with IL2/IL15-NKs (Figure 1C). In accordance, the supernatants of HCV-Huh7.5 cells previously exposed to IFNα-NKs could infect freshly plated Huh7.5 cells to a significantly smaller degree than the supernatants of cells co-cultured with IL2/IL15-NKs (Figure 1D). In addition, IFNα-NK treatment also led to ~90% reduction in intracellular HCV, as assessed using immunofluorescence staining for HCV-Ag (Figure 1E,F). Taken together, these results demonstrate that NK cells activated with IFN-α possess enhanced contact-dependent and -independent antiviral functions against HCV-infected cells.

### 3.2. IFNα-NKs Counteracted HCV Infection In Vivo More Efficiently Than IL2/IL15-NKs

To test the antiviral activity of IFNα-NKs in vivo, we used the chimeric uPA/SCID mice, which harbor human hepatocytes stably integrated in the liver parenchyma [19]. Mice were infected with HCV-JFH1 virions and were later treated with two doses of IL2/IL15-NKs or IFNα-NKs [30]. HCV viremia was traced weekly by performing blood quantification. While the insufficient number of mouse replicates (*n* = 2/group) in our study does not allow any unbiased conclusion to be drawn, we saw a trend of HCV infection at any time point considered, with the highest viremia being observed in those animals treated with IL2/IL15-NKs and the lowest in the IFNα-NK-injected group (Figure 2A).

Human albumin was used as a readout of both the chimerism rate in uPA/SCID mice and NK-cell cytotoxicity against human hepatocytes (Figure 2B). The infusion of NK cells did not affect human hepatocytes function, as the levels of human albumin fluctuated in the physiological range in all mouse groups (Figure 2B). Mice were sacrificed six weeks after the first NK-cell injection, and liver samples were collected for further analyses. The histological analyses revealed the presence of chimerism, with human hepatocytes (H) being neatly distinguishable within the mouse liver parenchyma (M), owing to a clear cytoplasm corresponding to glycogen storage and lipid droplets [19] (Figure 2C). Interestingly, we observed a reduced expression of HCV non-structural protein 3 (HCV NS3) in the livers of IFNα-NK-treated mice (Figure 2(Ciii)) as compared with IL2/IL15-NK-treated siblings (Figure 2(Cii)). Conversely, in both treated groups, liver stained positively for CD56, the archetypal phenotypic marker of human NK cells [32], suggesting that the IL-2/IL-15 activation protocol did not interfere with natural NK-cell homing to the liver (Figure 2(Cii,iii)) [5]. Finally, nested PCR for HCV negative-stranded RNA replicative intermediate indicated a lack of viral replication following IFNα-NK treatment but not after IL2/IL15-NK treatment (Figure 2D). These data confirm that IFN-α greatly enhances NK-cell antiviral responses in vivo.

### 3.3. IFNα-NKs Withstood HCV-Induced Functional Impairment, and Their Antiviral Effects Were Even Maintained after Therapy Withdrawal

HCV evades innate immune surveillance by impairing NK-cell function, thus establishing chronic infection [43,44,45]. To understand whether activation with cytokines could revert NK-cell inhibition and restore the immune response, NK cells were exposed to HCV virions for four hours and then stimulated with IL-2/IL-15 or IFN-α overnight. Using a transwell co-culture system with HCV-Huh7.5 cells, we quantified the levels of HCV-Ag, as we did previously. We found that HCV pre-treatment significantly reduced the antiviral efficacy of IL2/IL15-NKs, whereas IFNα-NKs function was barely compromised (Figure 3A). We wanted to dissect the dynamics and durability of NK-cell responses over time. First, we used Huh7.5 cells infected with the HCV-JFH1 replicon tagged with fluorescent reporter gene Venus and flow cytometry. HCV-Venus Huh7.5 target cells were co-cultured in transwells with either IFNα-NKs or IL2/IL15-NKs for 72 h. The transwell inserts were removed, and the target cells were kept in culture for three additional days (until Day 6). Where indicated, NK cells were exposed to HCV prior to activation with cytokines, to address the inhibitory role of the virus (Figure 3B and Appendix A). In the presence of IFNα-NKs, the number of HCV-Venus Huh7.5 cells dropped below 10% in 72 h. By contrast, IL2/IL15-NK treatment resulted in a smaller, yet significant, decrease in HCV-Venus Huh7.5 cells (Figure 3B). Importantly, the percentage of HCV-Venus Huh7.5 remained negligible after the removal of IFNα-NKs from the co-culture (Day 6), while it rebounded shortly in the IL2/IL15-NK group (Figure 3B). Importantly, NK cells that had been exposed to HCV before the start of the co-culture could still effectively counteract HCV infection when activated with IFN-α, whereas IL2/IL15-NK functions were impaired (Figure 3B).

Next, we compared the antiviral efficacy of IFNα-NKs with IFN-α monotherapy. IFNα-NKs or IFN-α alone were added to HCV-infected cells for seven days. The treatment was then removed, and the target cells were kept in culture for five additional days (until Day 12). HCV-Ag levels were monitored on Days 3, 7, and 12. HCV-Ag levels were only mildly affected during the first three days of IFNα-NK treatment, while they were almost undetectable 7 or 12 days later (Figure 3C). Treatment with IFN-α alone resulted in a faster response, with HCV-Ag levels alr ready reduced on the third day of treatment (Figure 3C). However, the antiviral response was strictly dependent on the presence of the cytokine in the medium, as HCV-Ag levels significantly rebounded following IFN-α removal (Figure 3C). To confirm the suppression of HCV replication at the molecular level, we checked, using nested PCR, for the presence of the negative-stranded RNA intermediate [46].

In line with our previous findings, we confirmed the absence of HCV replication on Day 12 in the presence of IFNα-NKs (Figure 3D). Thus, our findings highlight early relapses of HCV infection after discontinuation of IFN-α monotherapy and stable HCV suppression under IFNα-NK treatment.

### 3.4. Secretion of IFN-γ Contributed to IFNα-NK-Mediated Antiviral Effects

To identify the soluble factors released upon specific activation, NK cells were stimulated overnight with either IL-2/IL-15 or IFN-α. The expressions of cell death mediators and pro-inflammatory cytokines were investigated with flow cytometry and multiplex ELISA. Using flow cytometry, we found that IFNα-NKs had higher levels of TRAIL and IFN-γ, while IL2/IL15-NKs had increased levels of CD107a and TNF-α (Appendix A) [8,47,48]. Receptors associated with cytolytic activity such as NKp30, NKp44, NKp46, and NKG2D were expressed at comparable levels by both IL2/IL15- and IFNα-NKs (Appendix A).

Of 18 cytokines quantified in the NK-cell-conditioned media with multiplex ELISA, only soluble TRAIL (s-TRAIL) and IFN-γ were differentially released in IFNα-NKs and IL2/IL15-NKs (Figure 4A). Other cytokines, including perforin, granzyme A and B, macrophage inflammatory protein-1 beta (MIP-1β), sCD137, granulocyte-macrophage colony-stimulating factor (GM-CSF), IL-13, and soluble FAS ligand (sFASL) were released at similarly high levels by both populations (Appendix A). Likewise, interferon-γ induced protein 10 kDa (IP-10), RANTES, MIP-1α, FAS, and tumor necrosis factor alpha (TNF-α) were released at intermediate concentrations, while IL-4, IL-6, and IL-10 were similarly released at very low concentrations by both populations (Appendix A).

To investigate whether s-TRAIL and IFN-γ could contribute to the enhanced antiviral functions of IFNα-NKs, we co-cultured HCV-Huh7.5 cells in transwells with either IL2/IL15-NKs or IFNα-NKs and added neutralizing monoclonal antibodies to block the function of s-TRAIL or IFN-γ. As shown in Figure 4B, the neutralization of s-TRAIL did not affect the IL2/IL15-NK or IFNα-NK response, as indicated by unchanged HCV-Ag levels in the media. However, we could appreciate a significant increase in HCV-Huh7.5 cell viability, which was more pronounced following IFNα-NK treatment, suggesting a role in apoptosis for s-TRAIL (Appendix A) [49]. Conversely, when we neutralized IFN-γ, we observed a significant increase in HCV-Ag levels in the IFNα-NK group and no effects in the IL2/IL15-NK group (Figure 4C), which identifies IFN-γ as an important mediator of the IFNα-NK antiviral response.

### 3.5. Secretome of IFNα-NKs Was Enriched in TRAIL, OAS1, ISG15, and LGALS9

The neutralization of IFN-γ produced an increase in the percentage of infection of about 60% in HCV-Huh7.5 cells co-cultured with IFNα-NKs (Figure 4C). As the infection was not fully restored after the IFN-γ blockade, we postulated that other soluble factors could contribute to the suppression of HCV infection exerted by IFNα-NKs. To further characterize the mediators released by NK cells under different conditions, we used mass-spectrometry-based proteomics. Briefly, the secretome of NK cells, untreated or treated with IFN-α or IL-2/IL-15, was analyzed with tandem mass tag (TMT)-based quantitative proteomics and combined into a multiplex sample. The TMT multiplexed sample was split in two fractions (F1 and F2) and run on a nano–liquid chromatography system coupled with mass spectrometry (nLC-MS). Proteins were quantified, and their values were normalized to ensure equal protein amounts across replicates (Appendix A). A total of 23,931 peptides were identified in F1 and F2; after the removal of contaminants and missing values across all replicates, 3277 proteins were left (Appendix A). Of these, 190 proteins were classified as extracellular, according to the Gene Ontology cellular component terms. The distribution of peptide spectrum matches (PSMs) over the Δmass was below 5 ppm for at least 95% of identified peptides, confirming the high quality and resolution of our analysis (Appendix A).

Four extracellular proteins were consistently up-regulated at least three-fold in the secretome of IFNα-NKs as compared with that of unstimulated NK cells (Figure 5A). These were: (1) TNFSF10 or s-TRAIL, inducer of cell apoptosis [50]; (2) interferon-induced dsRNA-activated antiviral enzyme 2′-5′-oligoadenylate synthetase 1 (OAS1), which plays a critical role in the cellular innate antiviral response [51]; (3) the ubiquitin-like interferon-stimulated gene 15 kDa ISG15, which plays a key role in the host response to pathogens [52]; and (4) β-galactoside-binding protein galectin-9 (LSGAL9), which has immunomodulatory effects on several cell types [53]. These proteins were also among the most relevantly up-regulated in the whole proteome (Appendix A). Of note, not only were TRAIL, OAS1, ISG15, and LGALS9 the most up-regulated proteins in the IFNα-NK secretome (Figure 5A) but also the most highly expressed as compared with IL2/IL15-NKs (Figure 5B).

### 3.6. Galectin-9 and IFN-γ Released by IFNα-NKs Mediated Suppression of HCV Infection

Of the four key proteins that were identified with nLC-MS, s-TRAIL was already shown not to be directly involved in the NK-cell response to HCV in our system (Figure 4B). The close correlation of the free forms of ISG15 and OAS1 with IFN-γ during the antiviral response has been widely described in the literature [54]. Moreover, we demonstrated that the blockade of IFN-γ only partially inhibited the antiviral response of IFNα-NKs (Figure 4C). For these reasons, we decided to focus on LGALS9 (galectin-9).

First, we validated the proteomics data of galectin-9 with ELISA (Figure 5C) and Western blot (Appendix A) on the secretome of both IFNα-NKs and IL2/IL15-NKs. We also analyzed the cellular lysates with a Western blot and found galectin-9 in unstimulated cells (Appendix A). These data suggest that galectin-9 is contained in the cytosol [55] and that is released by NK cells following IFN-α stimulation. To investigate the role of galectin-9 in the antiviral response, we co-cultured HCV-Huh7.5 cells with IFNα-NKs in the presence or not of a blocking antibody specific for galectin-9. We found that HCV-Ag levels significantly increased when galectin-9 was neutralized (Figure 5D) [26]. Importantly, the percentage of infection further increased when both galectin-9 and IFN-γ were concomitantly neutralized (5D).

The effects of galectin-9 have been partly attributed to its interaction with the immunosuppressor T cell immunoglobulin and mucin protein 3 (TIM-3) receptor [56], although controversies exist regarding whether TIM-3 truly serves as a binding partner for galectin-9 both in mice and humans [57]. To understand whether TIM-3 might mediate galectin-9 effects, we first tracked the expressions of both galectin-9 and TIM-3 over time under either IL-2/IL-15 or IFN-α activation. In the steady state, NK cells showed the intracellular expression of galectin-9 and negligible levels of surface TIM-3 (Figure 5E,F and Appendix A). After 4 and 24 h stimulation with either IFN-α or IL-2/IL-15, the levels of galectin-9 were increased in NK cells, yet TIM-3 levels remained low (Figure 5E,F). Taken together, these results seem to suggest a non-involvement of TIM-3 in NK-cell antiviral responses in our system. Moreover, we observed a joint effect of galectin-9 and IFN-γ in suppressing the HCV infection cycle.

### 3.7. In Silico Prediction of Binding of HCV Envelope Proteins to Galectin-9 Carbohydrate-Recognition Domains

Galectins have high affinity for β-galactose-containing glycol-conjugates and contain carbohydrate-recognition domains (CRDs) at their C-terminal and N-terminal regions [58,59,60,61]. HCV contains two envelope glycoproteins, E1 and E2, which are involved in virus entry into the host cell [62]. We hypothesized that galectin-9 might bind to HCV E glycoproteins. To test this, we used the available structural data in the literature, searching for a complex between galectin-9 and HCV E glycoproteins. Unfortunately, no experimentally solved structures of the complex were available. Thus, we worked on the single interacting structure partners applying computational techniques to simulate putative binding modes between galectin-9 and HCV E glycoproteins. From the Protein Data Bank (PDB) database [35] only the E2 portion of the whole HCV E protein and the N-terminal region of galectin-9 were available. We adopted in silico approaches to specifically evaluate the binding mode, the residues possibly involved in the interaction, and the complex stability. In detail, we chose the best structures available based on organism (Homo sapiens for galectin-9), resolution, the completeness of the carbohydrate chains, and the variability of the carbohydrates bound to the E2 region. For HCV E2, PDB structures 6MEJ and 6MEK [34] were selected. For the galectin-9 CRDs, oligosaccharide-complexed structures were chosen (2ZHM and 2ZHN) [33]. Hence, two protein–protein docking screenings were run in parallel using PDB structures 2ZHM and 6MEK for one protein–protein docking, and 2ZHN and 6MEJ for the other one, without any supervised bias.

Contacts between the E2 oligosaccharides and galectin-9 key amino acids His61, Asn63, Arg65, Asn75, Trp82, Glu85, Arg87, and Asn137 were analyzed by selecting the best docking poses according to the literature, and the interactions were established between the galectin-9 CRDs and E2 oligosaccharides [33,63]. Noteworthy for both docking results is that the top-ranked poses reported very similar complexes, where a branched oligosaccharide of E2 accommodated into the binding region of the galectin-9 CRDs (Figure 6A).

In order to evaluate the complex stability and explore the most stable and frequent interactions between E2 oligosaccharides and the galectin-9 CRDs, we explored the contribution of amino acid contacts between protein partners. Our computational studies suggested that the sugar moieties of HCV E2 might not be the only ones responsible for contacting the galectin-9 CRDs (Figure 6B), as the amino acids belonging to HCV E2 established interactions with galectin-9 CRD residues (Appendix A). The two selected docking poses of the galectin-9 CRDs in complex with HCV E2 were submitted to molecular dynamics (MD) simulations (250 ns for each complex). The interaction stability and frequency of the observed interactions were investigated during the whole MD timeframe. Indeed, both complexes provided stable RMSD (root-mean-square deviation) plots during the entire trajectories (Appendix A). Furthermore, the interactions observed within the docking were maintained mostly stable during both MD simulations, showing an important role in the complex stabilization (Figure 6C).

The analysis of these results suggested that the complex involving PDB 2ZHN and 6MEJ registered a higher number of key interactions between the galectin-9 CRDs and the E2 branched oligosaccharide. Furthermore, in order to explore the contacts between galectin-9 and E2 amino acids, the two MD simulations were processed by clustering the MD frames in 10 clusters per trajectory [64]. The centroids of each cluster were thus analyzed to deepen the single amino acid role in the complex stabilization (Appendix A). As can be observed, some identified protein–protein contacts were shared by both complexes, even though in some cases they did not occur during both the entire simulations. The most frequent interactions between protein partner residues are listed in Appendix A. These encouraging in silico results prompted us to test in vitro interaction between galectin-9 and HCV using a co-immunoprecipitation assay. Only HCV virions that had previously been incubated with purified galectin-9 were trapped by the sepharose beads precoated with the anti-galectin-9 antibody (Figure 6D), thus confirming the interaction between the two proteins. 

To address a possible neutralizing function, we incubated HCV with increasing doses of galectin-9, and we tested infectivity on Huh7.5 cells. By performing immunofluorescence, we observed that HCV infectivity was inversely proportional to the dose of galectin-9 added, further suggesting that galectin-9 might work as a neutralizing factor of HCV infection (Figure 6E,F).

### 3.8. IFIT1 and MX1 Were Up-Regulated in Target Cells Following IFNα-NK Treatment

We next performed a comparative proteomic profiling of cell extracts of HCV-Huh7.5 target cells after exposure to either IL2/IL15-NKs or IFNα-NKs. Two proteins, namely, interferon-induced protein with tetratricopeptide repeats 1 (IFIT1) and myxovirus-1 (MX1), appeared to be dysregulated in opposite directions, with IFNα-NKs showing a >3-fold increase and IL2/IL15-NKs showing reduced levels (Figure 7A). Both proteins are involved in the antiviral response mounted by host cells and are dependent on IFN signaling [65]. In particular, IFIT1 inhibits HCV replication by blocking internal ribosome entry site (IRES)-dependent translation [66]. The MX1 protein is a GTPase that acts in the early step of the viral life cycle by blocking the endocytic traffic of incoming virus particles and preventing them from reaching their cellular destination [67,68]. We decided to focus on MX1. By conducting a Western blot, we confirmed that the MX1 expression was induced in HCV-Huh7.5 cells by IFN-γ in a dose-dependent manner, but not by galectin-9, (Figure 7B and Appendix A). The lack of responsiveness to galectin-9 might be due to negligible levels of TIM-S3 in Huh7.5 cells, as it was reported by others [56] and further confirmed by us (Appendix A). When target cells were incubated in transwells with NK cells, we found the upregulation of MX1 only in the presence of IFNα-NK and not with IL-2/IL-15-NK cells (Figure 7C and Appendix A). Taken together, our data highlight MX1 as a downstream target of IFN-γ signaling in HCV-Huh7.5 cells.

## 4. Discussion

HCV infection accounts for more than 70 million cases around the world and is thus recognized as a major health problem. In recent years, a tremendous effort has been made to establish experimental models that can recapitulate the clinical manifestations of CHC and help to expand the number of therapeutic options. The chimpanzee, which is the only non-human primate susceptible to HCV infection and also the only suitable animal model for the dissection of the innate and adaptive immune responses to HCV, has almost been abandoned due to ethical, economic, and practical constraints. These limitations have led to the establishment of humanized mouse models that can support HCV research by allowing the engraftment of human hepatocytes to be performed without evidence of rejection [69]. Yet, these models, including the uPA/SCID mice that we used in this study, are difficult to breed due to lethal genetic modifications [70] and are costly [71].

Besides in vivo models, the HCV cell culture system that exploits Huh7.5 cells transfected with the HCV-JFH1 genome provides a useful means for the investigation of HCV infection [17]. Huh-7.5 cells, which are a clone of the human hepatoma Huh7 cell line, support HCV propagation well, owing to a mutation in the retinoic-acid-inducible gene-I (RIG-I), which impairs IFN signaling [72]. Using this approach, we investigated the role of NK cells as antiviral effectors, identifying key molecules that might be considered to suppress viral infection. We confirmed that NK-cell function could be significantly enhanced in vitro following exposure to IFN-α. Compared with IL-2/IL-15-based activation, IFN-α endowed NK cells with greater contact-dependent and -independent responses against HCV-infected cells in vitro. The enhanced function of IFNα-NKs might result from their ability to withstand the inhibiting effects of HCV. Importantly, IFNα-NKs also appeared to better counteract HCV infection in vivo. We are aware of the weakness of our in vivo study, where only two mice per group were used. However, our main aim was not to investigate IFNα-NK effector functions in vivo but to understand the mechanisms of the IFNα-NK-induced suppression of HCV.

The encouraging findings prompted us to investigate the dynamics, durability, and mechanisms of action of IFNα-NK-mediated HCV inhibition in vitro. We found that IFNα-NKs reduced the number of HCV-infected cells below 10% within 72 h, while IL2/IL15-NK treatment was associated with a smaller reduction. Importantly, reduced numbers of HCV-infected cells were maintained after the discontinuation of IFNα-NK treatment, while they rebounded shortly after the discontinuation of IL2/IL15-NK treatment. Moreover, IFNα-NK treatment was found to be superior to standard IFN-α monotherapy. Long-term follow-up studies are needed to confirm the enhanced stability of IFNα-NK treatment over other regimens.

At the molecular level, IFNα-NKs were characterized by high levels of membrane-bound and -secreted TRAIL and IFN-γ. Interestingly, s-TRAIL was dispensable for NK-cell antiviral functions, while the blockage of IFN-γ reduced the ability of IFNα-NKs to suppress HCV infection. Through MS-based proteomics, we found that IFNα-NKs up-regulated four proteins as compared with IL2/IL15-NKs: TRAIL, OAS1, ISG15, and LGALS9. We decided to focus on LGALS9 (galectin-9), and we showed that the neutralization of this factor could hinder the IFNα-NK-mediated antiviral response to the same extent of the IFN-γ blockage. Importantly, the concomitant neutralization of both galectin-9 and IFN-γ significantly suppressed the antiviral response of IFNα-NKs and restored the infection. While other studies are needed to understand whether galectin-9 interacts with a receptor on target cells or not, we have evidence that gives us reason to believe that it might help to contain HCV infection through its binding to and neutralization of HCV virions. First, we predicted the binding of HCV envelope protein E2 to galectin-9 carbohydrate-recognition domains. Next, by performing co-immunoprecipitation assays, we confirmed physical interaction. We speculate that this interaction might reduce the ligation of NK-cell signaling molecules such as CD81 by viral proteins [73], which would explain the refractoriness of IFNα-NKs to the functional impairment induced by HCV (Figure 8).

Finally, we leveraged proteomics to investigate the effects of NK-cell treatment in HCV-Huh7.5 target cells at the molecular level. We found that target cells up-regulated IFIT1 and MX1 following IFNα-NK treatment and that MX1 expression was induced by IFN-γ but not by galectin-9 (Figure 8).

Our data demonstrated the important role of IFN-α-activated NK cells in the control of HCV infection using a robust and reproducible cell culture system. Giving that NK cells respond against both infected and tumor cells, it is worth assessing the benefits of IFNα-NK therapy in the context of other tumor viruses for which a pharmacological cure is not yet available, such as the Epstein–Barr virus (EBV), human immunodeficiency virus (HIV), hepatitis B virus (HBV), and cytomegalovirus (CMV) [74].

Cell-free derivatives from NK cells are also being considered for cancer and viral immunotherapy, and our study highlights two main mediators of IFNα-NK antiviral response that could be exploited for treatment: IFN-γ and galectin-9. IFN-γ is involved in the antiviral response, and the mechanisms by which it suppresses infection have been largely investigated [75]. Galectin-9 is a pleiotropic immune modulator affecting numerous cell types of innate and adaptive immunity.

The participation of galectin-9 and other galectin family members in virus infection is well accepted [76,77,78]; however, whether galectins are friends or foes is not clear yet. The expression of galectin-9 was reported in hepatocytes and Kupffer cells from the biopsies of HCV patients [78], and the levels of circulating galectin-9 and galectin-9-expressing Tregs [79] in the sera of HCV patients [80] correlated positively with the persistence of HCV infection and the progression of chronic liver disease [81]. Both cytokine activation protocols resulted in comparable intracellular levels of galectin-9. However, the secretion of galectin-9 was promoted by IFN-α and not by IL-2/IL-15. At this moment, we are unable to explain such different outcome. The mechanism by which galectin-9 is secreted is complex, involving non-classical secretory pathways, and remains poorly understood [82]. Unlocking the secretory mechanisms of galectin-9 in our system would definitely pave the way for new strategies to manipulate galectin-9 expression and reinforce NK-cell-mediated response. However, studies into the regulation of galectin-9 secretion are complex, often requiring the use of drugs to stimulate or inhibit its secretion. Given off-target drug effects, conclusions might not be so straightforward.

The secretion of galectin-9 by IFNα-NKs endowed them with greater ability to suppress HCV infection. This beneficial role of galectin-9 disagrees with reports in which galectin-9 expression correlated with impaired NK-cell cytotoxicity and cytokine production [83,84,85]. It was reported that it is through interaction with TIM-3 that galectin-9 mediates its functions. The effects of this interaction with regards to NK-cell function remain, however, poorly understood. TIM-3 expression was correlated with NK-cell exhaustion in some studies [86], but there is also evidence to suggest that TIM-3 might be a positive regulator of NK-cell function through the enhancement of IFN-γ production in response to galectin-9 [87]. Our results seem to suggest a non-involvement of TIM-3 in the NK-cell antiviral responses in our system. In the steady state, healthy NK cells lack the expression of exhaustion markers such as TIM-3, which, by contrast, is upregulated in the NK cells of patients with CHC or advanced hepatocarcinoma [88]. NK-cell exhaustion, and impaired cytotoxicity and function may contribute to viral persistence, the chronicity of viral infection, and disease progression. Reverting NK-cell exhaustion through activation with cytokines such as IFN-α might represent an interesting approach to restore natural immunity.

As most of our knowledge concerning the process of galectin-9 secretion and cell phenotypic responses to galectin-9 derives from studies performed in vitro using recombinant proteins, we cannot exclude dose-dependent and divergent effects of galectin-9. Moreover, divergent in vitro/in vivo responses to galectin-9 are worth considering, given the supra-physiological concentrations often used in in vitro studies. We here demonstrated that even low concentrations of galectin-9 (25 ng/mL) could reduce viral infection in vitro. In the future, we aim to confirm galectin-9 as an important mediator of IFNα-NK antiviral responses in vivo using galectin-9 conditional knockout mice. Moreover, to establish galectin-9 as a potential drug to fight infection, we aim to investigate the effects of its cross-linking with other antigens and immune cell types in vivo.

## 5. Patents

NK-mediated immunotherapy and uses thereofInventor: Ester BadamiDate of publication: 24 October 2019Patent office: USPatent number 16464823

Description: The present invention refers to a method for the production of activated CD3-CD56+ NK cells; activated CD3-CD56 NK+ cells obtainable with the method; and their use, in particular for the treatment of a tumor (preferably a hepatocellular carcinoma (HCC)), for use in the treatment and/or prevention of an HCV infection, for use in the treatment and/or prevention of a post-liver transplant HCV reinfection, or for use for the prevention of a post-liver transplant HCC recurrence. The invention also concerns pharmaceutical compositions including activated CD3-CD56+ NK cells.

## Figures and Tables

**Figure 1 viruses-14-01538-f001:**
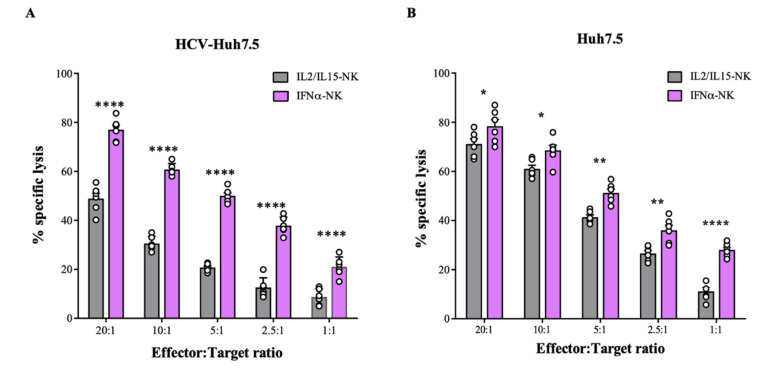
Functional characterization of NK cells activated with IL-2/IL-15 or IFN-α. (**A**,**B**) Percentage of lysed target HCV-Hu7.5 (**A**) or Huh7.5 (**B**) cells following exposure to decreasing ratios of IL2/IL15-NKs (gray bars) or IFNα-NKs (lilac bars), as calculated with a four-hour chromium release assay. Each dot represents the average of three technical replicates from one donor (*n* = 6). Mean values ± SEM are shown. (**C**) Infection rates (percentage of control) of HCV-Huh7.5 cells exposed to IL2/IL15-NKs or IFNα-NKs using a transwell co-culture system. Infection rates were calculated by measuring HCV-Ag levels released in the culture media seven days after co-culture. Each dot represents the average of three technical replicates from one donor (*n* = 10). Mean values ± SEM are shown. (**D**) Infection rates (percentage of control) of freshly plated Huh7.5 cells following exposure to medium conditioned with HCV-Huh7.5 cells that were co-cultured for seven days with IL2/IL15-NKs (gray bar) or IFNα-NKs (lilac bar). Each dot represents the average of three technical replicates from one donor (*n* = 4). Mean values ± SEM are shown. (**E**) Representative immunofluorescence staining of HCV-Ag (green color) in HCV-Huh7.5 cells seven days after co-culture with IL2/IL15-NKs (**middle image**) or IFNα-NKs (**right image**). Untreated cells were used as controls (**left image**). DAPI (blue color) was used to stain the nuclei. Magnification of 200×. (**F**) Intensity of fluorescent HCV-Ag signal in cells as in (**E**), expressed as arbitrary units (AU), as quantified with ImageJ software. Each dot represents the average of three technical replicates from one donor (*n* = 5). Mean values ± SD are shown. Statistical analyses were performed using a two-way ANOVA with the Šídák multiple comparison test (**A**,**B**), unpaired Student’s *t*-test (**C**,**D**), and the one-way ANOVA with Tukey’s multiple comparisons test. * *p* < 0.05, ** *p* < 0.01, **** *p* < 0.0001; n.s., non-significant.

**Figure 2 viruses-14-01538-f002:**
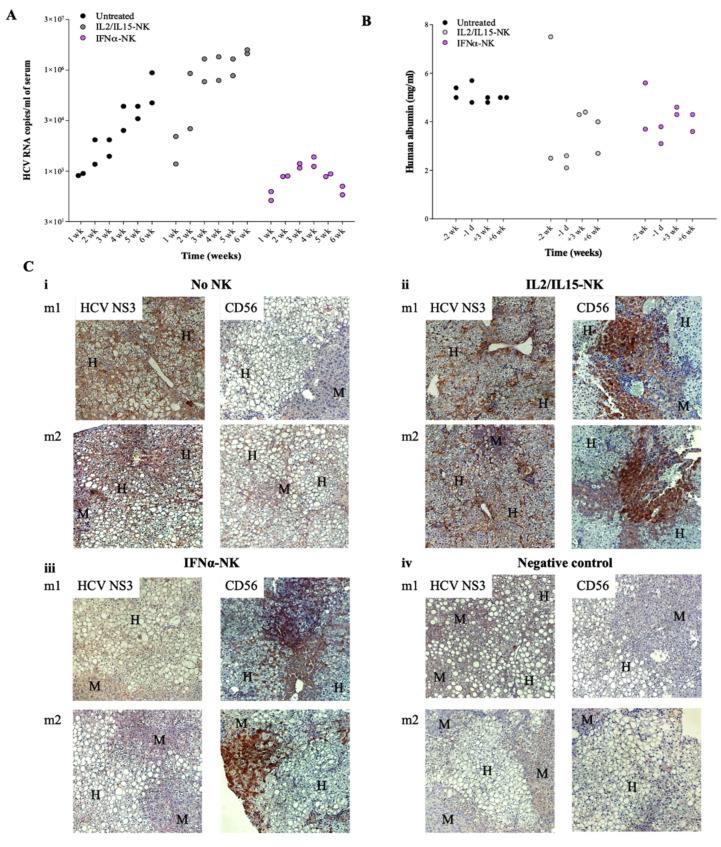
NK cells activated with IFN-α inhibited HCV replication in vivo. (**A**) Weekly HCV viremia levels in human-hepatocyte-engrafted and HCV-infected uPA/SCID mice following treatment with two doses of either IFNα-NKs (lilac dots) or IL2/IL15-NKs (gray dots) or no treatment at all (black dot) *(n* = 2/group). Each dot represents a mouse. (**B**) Concentrations (mg/mL) of human albumin in mice two weeks (−2 wk) or one day (−1 d) before NK-cell treatment and three (+3 wk) or six (+6 wk) weeks after NK-cell treatment. Each dot represents a mouse. (**C**) Representative images of immunohistochemical staining of either HCV NS3 or CD56 (brown color) in chimeric livers of untreated mice (**i**), or IL2/IL15-NK- (**ii**) or IFNα-NK-treated (**iii**) mice. Uninfected and untreated mice were used as negative controls (**iv**). Mayer’s hematoxylin (blue/purple color) was used to counterstain the nuclei, facilitating the identification of human hepatocytes (H) within the mouse (M) parenchyma. Magnification of 200×. (**D**) Representative agarose gel electrophoresis image showing the presence or absence of the HCV intermediate replication product (negative-strand RNA; 150 bp) in mouse livers as in **C** (m1 and m2 indicate mouse 1 and mouse 2, respectively).

**Figure 3 viruses-14-01538-f003:**
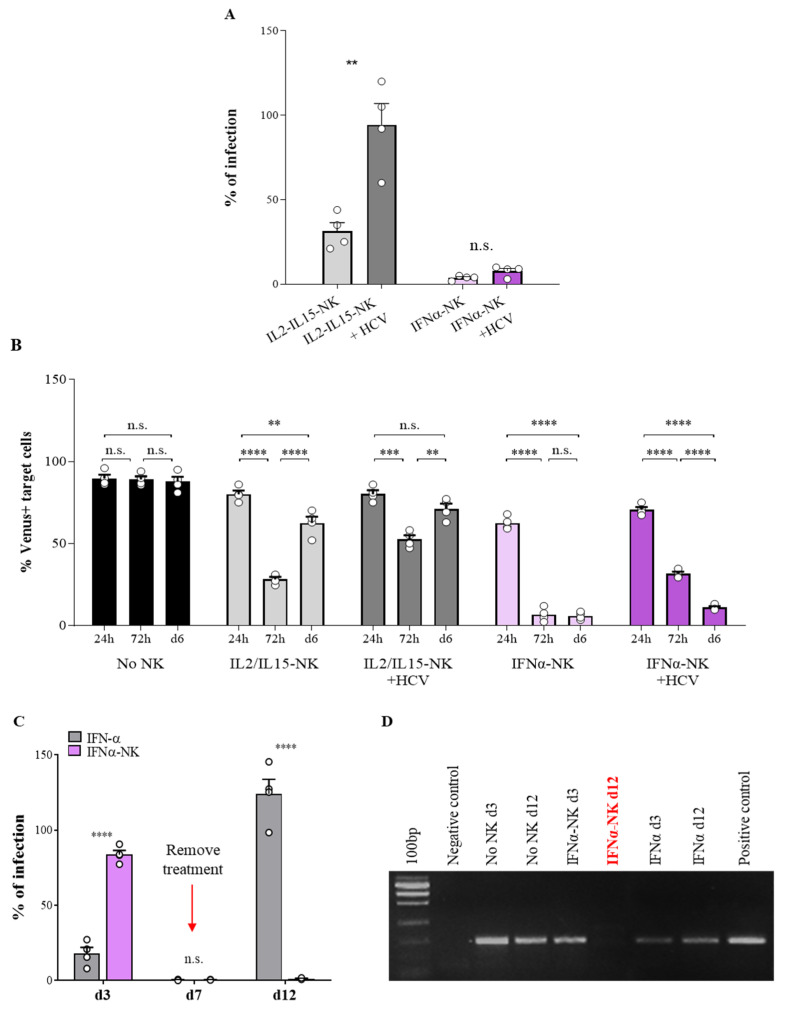
IFNα-NKs resisted to HCV-induced inhibition and showed durable antiviral responses in vitro. (**A**) Infection rates (percentage of control) of HCV-Huh7.5 cells co-cultured in transwells with IL2/IL15-NKs (gray bars) or IFNα-NKs (lilac bars) that had been previously exposed or not to recombinant HCV for four hours (darker gray or lilac bars). Each dot represents the average of three technical replicates from one donor (*n* = 4). Mean values ± SD are shown. (**B**) Percentages of HCV-infected (Venus+) Huh7.5 target cells in co-culture with IL2/IL15-NKs (gray bars) or IFNα-NKs (lilac bars) that had been previously exposed or not to HCV (darker gray or lilac bars). Untreated target cells were used as controls (black bars). Briefly, transwells with NK cells were removed 72 h after the start of the experiment and target cells were maintained in culture in fresh medium until Day 6. Fluorescence intensities of Venus were recorded at 24 h, 72 h, and on Day 6 using flow cytometry. Each dot represents the average of three technical replicates from one donor (*n* = 4). Mean values ± SEM are shown. (**C**) Infection rates (percentage of control) of HCV-Huh7.5 either co-cultured with IFNα-NKs (lilac bars) or exposed to 1 µg/mL recombinant IFN-α (gray bars), as calculated by measuring HCV-Ag levels on Days 3, 7, and 12. On Day 7, transwells with IFNα-NKs were re-moved, and the medium was replaced in all wells. Each dot represents the average of three technical replicates from one donor (*n* = 4). Mean values ± SEM are shown. (**D**) Representative agarose gel electrophoresis image showing the presence or absence of the HCV intermediate replication product (negative-strand RNA; 150 bp) in HCV-Huh7.5 cells treated as in **C**. Untreated target cells were used as internal controls. Negative and positive PCR controls were also included in the reaction. Data are representative of four separate experiments performed using NK cells from four different donors. Statistical analyses were performed using unpaired Student’s *t*-test (**A**,**C**) and the two-way ANOVA with Šídák multiple comparison test (**B**). ** *p* < 0.01, **** *p* < 0.0001; n.s., non-significant.

**Figure 4 viruses-14-01538-f004:**
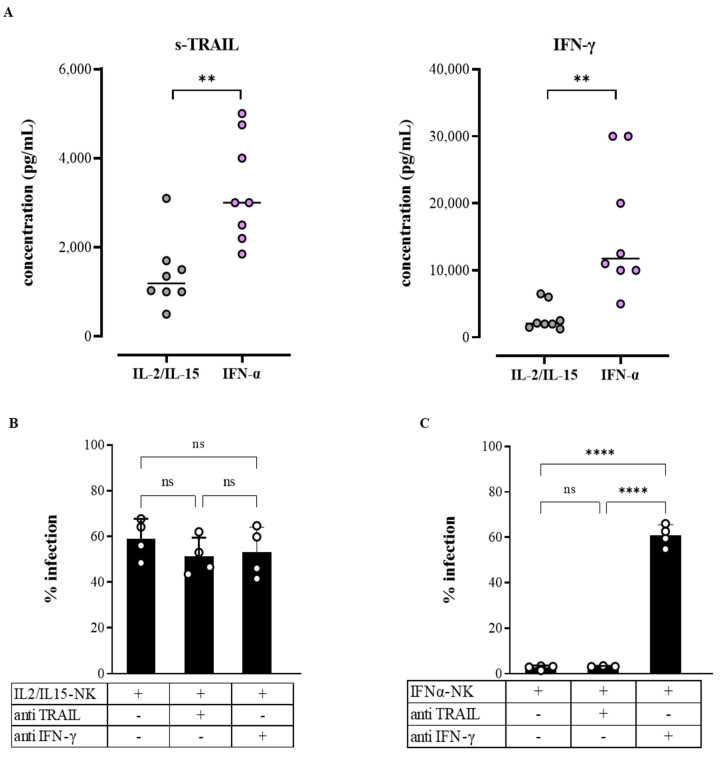
s-TRAIL and IFN-γ were differentially released by IFNα-NKs and IL2/IL15-NKs, but only IFN-γ contributed to the enhanced antiviral functions of IFNα-NKs. (**A**) Dot plots of s-TRAIL and IFN-γ cytokine expression (pg/mL) in the conditioned media of NK cells activated overnight with IL-2/IL-15 (gray) or IFN-α (lilac). Each dot represents a donor (*n* = 8). Means are shown. (**B**,**C**) Infection rates (percentage of control) of HCV-Huh7.5 cells exposed in transwells to either IL2/IL15-NKs (**B**) or IFNα-NKs (**C**). Infection rates were calculated by measuring HCV-Ag levels released in the culture media seven days after co-culture. Where indicated, a blocking anti-TRAIL antibody (10 µg/mL) or an anti-IFN-γ neutralizing antibody (10 µg/mL) was added to the culture. Each dot represents the average of three technical replicates from one donor (*n* = 4). Mean values ± SEM are shown. Statistical analyses were performed using Student’s *t*-test (**A**) and the one-way ANOVA with Tukey’s multiple comparisons test. ** *p* < 0.01, **** *p* < 0.0001; n.s., non-significant.

**Figure 5 viruses-14-01538-f005:**
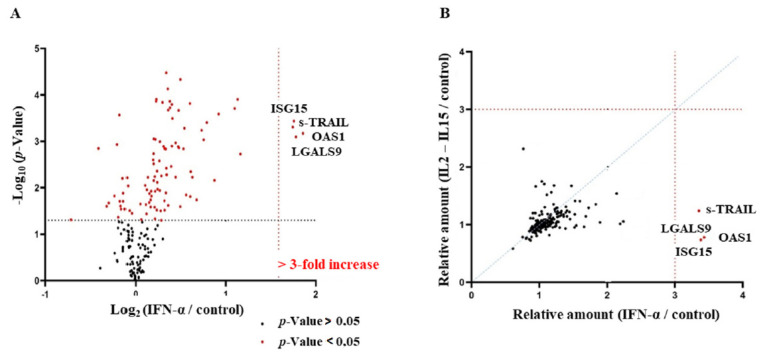
Secretion of galectin-9 and IFN-γ mediated HCV suppression by IFNα-NKs. (**A**) The volcano plot displays extracellular proteins (*n* = 190) differentially expressed in the secretome of IFNα-NKs as compared with the secretome of untreated NK cells, as identified and quantified with MS-based quantitative proteomics. Plotted along the *x*-axis is the mean of log2 fold-change; along the *y*-axis is the negative logarithm to the base 10 of *p*-values from Student’s t-test. The vertical black dashed line and the horizontal line reflect the filtering criteria (fold change ≥ 2.0 and *p*-values ≤ 0.05). Significant hits are depicted in red. The annotated dots are the four data points that have the largest distance from the origin and are above the fold change ≥ 3.0 (vertical red dashed line) and *p*-values ≤ 0.05 thresholds. (**B**) Correlation between secretome data of IFNα-NKs (*x*-axis) and IL2/IL15-NKs (*y*-axis). Relative expression levels as determined with MS-based quantitative proteomics are plotted for 190 proteins. Linear regression is indicated. Red dashed lines reflect fold change ≥ 3.0. The annotated red dots represent four proteins that were over-3-fold upregulated in the IFNα-NK secretome. (**C**) Levels of galectin-9 (pg/mL) in the conditioned media of NK cells from different donors (*n* = 11) activated with IL2/IL15-NKs (gray bars) or IFNα-NKs (lilac bars), as determined with the ELISA. Mean values ± SEM are shown. (**D**) Infection rates (percentage of control) of HCV-Huh7.5 cells exposed in transwells to IFNα-NKs. Infection rates were calculated by measuring HCV-Ag levels released in the culture media seven days after co-culture. Where indicated, anti-galectin-9 neutralizing antibody (10 µg/mL) and anti-IFN-γ neutralizing antibody (10 µg/mL) were added. Each dot represents the average of three technical replicates from one donor (*n* = 4). Mean values ± SEM are shown. (**E**,**F**) Mean fluorescence intensity (MFI) of intracellular galectin-9 and TIM-3 (**F**) signals in CD3^-^CD56^+^ NK cells as calculated with flow cytometry in the steady state and 4 or 24 h after stimulation with either IL-2/IL-15 or IFN-α. Each dot represents a technical replicate. Mean values ± SD are shown. Statistical analyses were performed using unpaired Student’s *t*-test (**C**,**E**,**F**) and the one-way ANOVA with Tukey’s multiple comparisons test (**D**). ** *p* < 0.01, *** *p* < 0.001, **** *p* < 0.0001.

**Figure 6 viruses-14-01538-f006:**
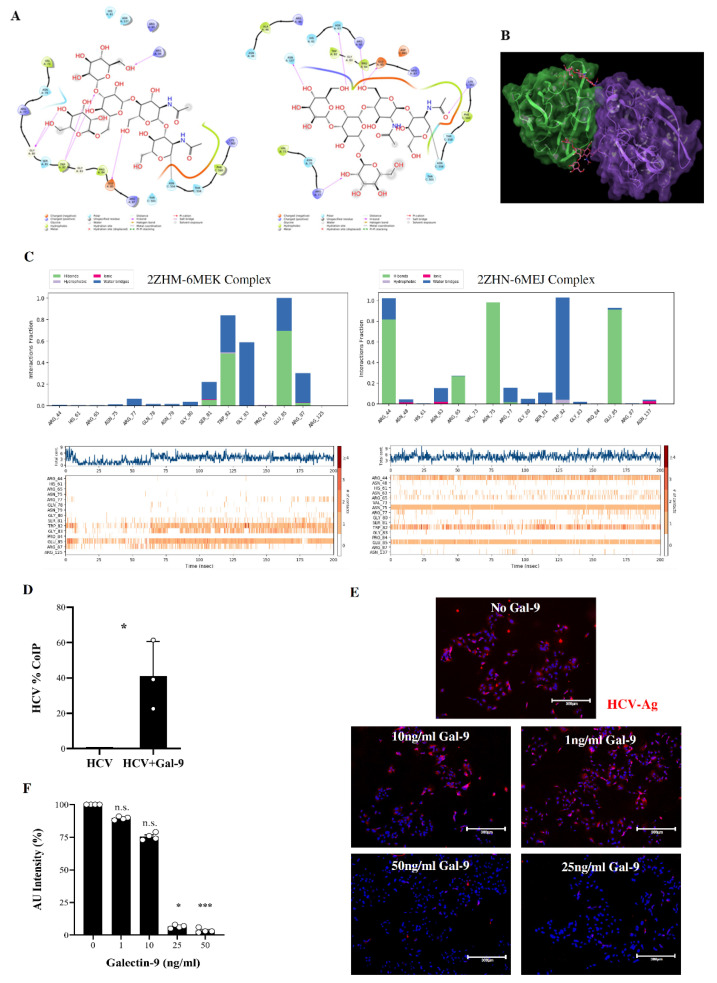
In silico and functional characterization of neutralizing interactions between HCV and galectin-9. (**A**) On the left, ligand interaction diagram of HCV E2 carbohydrate from PDB 6MEJ and galectin-9 CRDs from PDB 2ZHM; on the right, ligand interaction diagram of HCV E2 carbohydrate from PDB 6MEK and galectin-9 CRDs from PDB 2ZHN. (**B**) Galectin-9 CRDs–HCV E2 complex retrieved from a selected docking pose. The green chain is the galectin-9 CRDs from PDB 2ZHN, the purple chain represents HCV E2 from PDB 6MEJ, and the pink molecules are the sugar moieties belonging to E2. (**C**) On the top, bar plots of the interactions established between galectin-9 CRDs and HCV E2 during the MD trajectories; on the bottom, frequency occurrence of the interactions between the protein partners in the MD timeframe. (**D**) Bar graph showing the percentage of HCV bound to galectin-9, as assessed with the co-immunoprecipitation (CoIP) assay. HCV without galectin-9 was used as a negative control. Each dot represents a technical replicate. Statistical analyses were performed using Student’s *t*-test (*p*-value * ≤ 0.05). (**E**) Representative immunofluorescence staining of HCV-Ag (red color) in Huh7.5 cells infected with recombinant HCV preincubated with decreasing doses (50–0 ng/mL) of galectin-9. DAPI (blue color) was used to counterstain the nuclei. Magnification of 200×. (**F**) Intensity of fluorescent HCV-Ag signal in cells as in A, expressed as percentage of arbitrary units (AU), as quantified with ImageJ software. Each dot represents an independent experiment (*n* = 4). Mean values ± SEM are shown. Statistical analyses were performed using the Kruskal–Wallis test followed by Dunn’s multiple comparison test. * *p* < 0.05, *** *p* < 0.001; n.s., non-significant.

**Figure 7 viruses-14-01538-f007:**
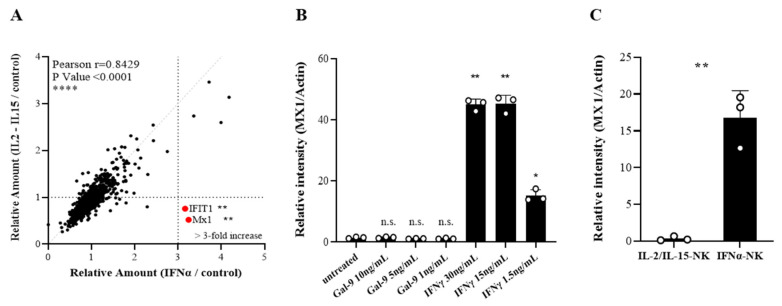
(**A**) Correlation between secretome data of HCV-Huh7.5 cells co-cultured with IFNα-NKs (*x*-axis) or IL2/IL15-NKs (*y*-axis). Relative expression levels, as determined with MS-based quantitative proteomics, are plotted for 2038 of proteins. Linear regression and Pearson’s r are indicated. The vertical line reflects a fold change ≥ 3.0 in IFNα-NKs with respect to IL2/IL15-NKs. The annotated red dots represent two proteins that were dysregulated in opposite directions. (**B**,**C**) Relative optical density of MX1 protein bands from Western blot films loaded with total protein extracts from HCV-Huh7.5 cells treated with decreasing doses of galectin-9 (1–10 ng/mL) or IFN-γ (1.5–30 ng/mL) (**B**) or stimulated with either IL2/IL15-NK or IFNα-NKs (**C**). Each dot represents a technical replicate. Mean values ± SD are shown. Statistical analyses were performed with the one-way ANOVA with the Holm–Šídák multiple comparisons test (**B**) or Student’s t-test (**C**). * *p* < 0.05, ** *p* < 0.01; n.s., non-significant.

**Figure 8 viruses-14-01538-f008:**
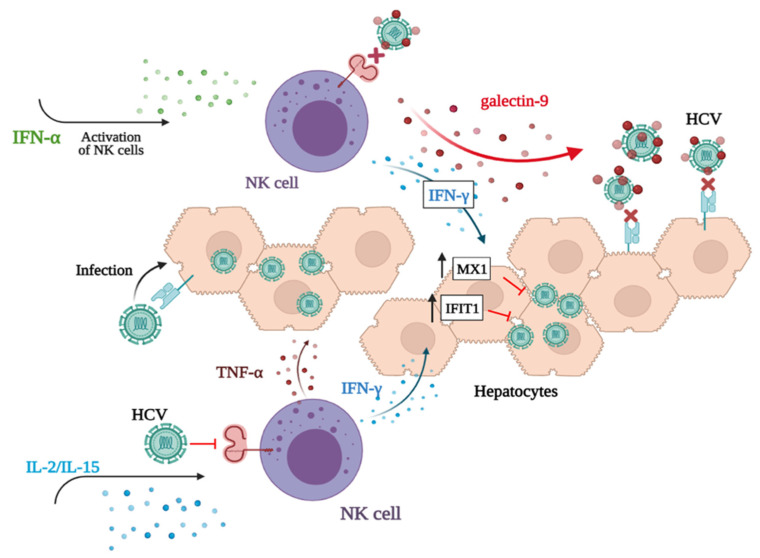
Cartoon depicting the response of NK cells to HCV infection following IL-2/IL-15 or IFN-α activation. The activation of NK cells with IFN-α induces the secretion of IFN-γ and galectin-9. IFN-γ drives the up-regulation of MX1 and IFIT1 by target hepatocytes, which contain/inhibit HCV replication. Extracellularly, Galectin-9 reduces HCV infectivity, putatively through its binding to viral-surface glycoprotein. This binding not only likely prevents the re-infection of target cells but also inhibits HCV ligation to receptors on NK-cell surface, thus interfering with the impairment of effector cells by the virus. The activation of NK cells with IL-2/IL-15 primarily induces the release of antiviral cytokines TNF-α and IFN-γ and only partially contains HCV infection. Galectin-9 is not secreted by NK cells upon activation with IL-2/IL-15. Figure was created with BioRender.com.

## Data Availability

Data are available in a publicly accessible repository. The data presented in this study are openly available in FigShare.

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
