# Peer review of "Galectin-9 and Interferon-Gamma Are Released by Natural Killer Cells upon Activation with Interferon-Alpha and Orchestrate the Suppression of Hepatitis C Virus Infection"

_viruses, 2022, doi:10.3390/v14071538_

Round 1
Reviewer 1 Report
Summary:
The authors present a study focusing on the antiviral effector functions of human primary NK cells in the context of HCV infection. Two different stimulation of NK cells (IFN-a and IL-2/IL-15) yielded different phenotypes regarding their potential to kill HCV-infected Huh7.5 cell line. Interestingly, priming NK cells with IFN-a strongly increase the amplitude and robustness of HCV-infected cell killing via the secretion of IFN-g and galectin-9. Although the study is well written and constructed, a couple of experiments should be added to finalize the mechanism proposed by the authors concerning the NK phenotype mediated by galectin-9 and IFN-g.
Major comments:
- Huh7.5 have an altered innate immune signaling pathway, namely RIG-I is mutated which results in poor activation at best. Consequently, the cell line is popular due to its permissiveness to HCV replication to propagate the virus or screen DAAs. However, when studying innate immunity and host-pathogens interaction, the relevance of this cell line is vastly diminished and the parental Huh 7.0 cell line would have been more appropriated. Especially the signaling from the infected Huh7.5 to NK cells would be very altered here. Nevertheless, that signaling is not the focus of the manuscript and the authors can address that in the discussion. I would suggest considering Huh7.0 in future work though.
- Figure 2: What is the rationale for only choosing 2 mice per conditions (Untreated, IL2/IL15-NKs or IFNα-NKs) knowing that the results analysis would be statistically problematic?
- Figure 3A: How long are the NK cells incubated with HCV virus? Are these NK cells supposed to get infected for the downregulation of the cell-killing phenotype? IDN-a treatment/activation of NK cells would naturally block viral infection and counteract any phenotype associated. Authors should clarify the timeline and provide some more controls with respect to the state of these NK cells (negative strand qPCR for instance).
- Effector to Target ratios should be stipulated in the figure legends for all experiments or in the method section if kept constant.
- Line 358: Please avoid using “(data not shown)”. If the data is not presented in the manuscript, the data should not be mentioned it in the results.
- If s-TRAIL is so induced in IFN-a treated NK cells (Figure 5A/ 5B), how do the authors explain that the treatment with neutralizing antibody did not yield any effect (Figure 4B/ 4C)?
- Figure 6A/B: Authors should validate the interaction between galectin-9 and HCV viral particle. Can the authors pellet galectin-9 when pelleting the virus by ultracentrifugation? Or using PEG? Be sure to include a negative control of Galectin-9 along in the media.
- Figure 6C/D: Does co-culture of IFN-a treated NK cells with Huh7.5 cells trigger induction of MX1 and IFIT1 in comparison to co-culture with unstimulated or IL-2/IL-15 treated NK cells? It is important to validate that phenotype to confirm this is part of the response at play here.
- The authors often use t-tests in panels where they have more than 2 conditions. This is not the correct statistical test. When they compare more than two conditions, they should use 1- or 2-way ANOVA followed by a post-hoc pairwise test corrected for multiple comparisons such as Tukey or Bonferroni tests.
Minor comments:
- Line 278: It is unclear if we are supposed to see the lipid droplets (LDs) in the histological work presented. Right now, the pictures are not of high quality enough to see intracellular details. Arrows indicating LDs to help the reader would be welcome.
- Supplemental Figure 4A: Authors should keep these histograms format consistent with the other (Grey and color overlap).
- Supplemental Figure 4B: Graphs appear to be pretty low quality; I can barely read the information on the axis.
- Supplemental Figure 4C: 3rd graph on the second line does not have the same y-axis as all the other graphs (Count instead of Count %). What is the meaning of count %?
Author Response
MAJOR POINTS
Point 1: Huh7.5 have an altered innate immune signaling pathway, namely RIG-I is mutated which results in poor activation at best. Consequently, the cell line is popular due to its permissiveness to HCV replication to propagate the virus or screen DAAs. However, when studying innate immunity and host-pathogens interaction, the relevance of this cell line is vastly diminished and the parental Huh 7.0 cell line would have been more appropriated. Especially the signaling from the infected Huh7.5 to NK cells would be very altered here. Nevertheless, that signaling is not the focus of the manuscript and the authors can address that in the discussion. I would suggest considering Huh7.0 in future work though.
Response 1: We thank the reviewer for the suggestion. We are aware that the Huh-7.5 cell line has limitations, however, it is acknowledged as a more permissive cell line for HCV propagation than its parental Huh 7.0 cell line (PMID: 15708988). This encouraged us to choose Huh7.5 cells for our experiments. We have now mentioned the altered RIG-I pathway in these cells in the discussion.
Point 2: Figure 2: What is the rationale for only choosing 2 mice per conditions (Untreated, IL2/IL15-NKs or IFNα-NKs) knowing that the results analysis would be statistically problematic?
Response 2: We agree with the reviewer that our in vivo study suffers from a lack of statistical power with few biological replicates. We have mentioned this in our manuscript. It was not our intention to use such a small number of animals/group. Indeed, we had commissioned a larger study to a laboratory that had the uPA/SCID mouse line and the expertise to perform the experiments. However, in the end, this laboratory could not fulfill our request, possibly due to a difficulty in breeding these animals, as we have discussed in our manuscript. For financial and logistic reasons, we could not later repeat our animal study on a larger cohort.
Point 3: Figure 3A: How long are the NK cells incubated with HCV virus? Are these NK cells supposed to get infected for the downregulation of the cell-killing phenotype? IDN-a treatment/activation of NK cells would naturally block viral infection and counteract any phenotype associated. Authors should clarify the timeline and provide some more controls with respect to the state of these NK cells (negative strand qPCR for instance).
Response 3: Preliminary experiments, where both a 4-hour and an overnight NK cell exposure to HCV-JFH1 were carried out before NK cell cytokine induced-activation, indicated similar effects on NK cells, which prompted us to use the shortest (i.e. 4-hour) incubation time. We have clarified this in our manuscript.
To the best of our knowledge, HCV productively infects hepatocytes and can be found in extrahepatic reservoirs (for example in the gut, or in peripheral mononuclear cells); yet whether this infection is productive or not is still matter of debate.
Addressing whether NK cells become infected with HCV in our system is beyond the scope of our study. The NK cell functional impairment that we observe after HCV exposure might be due to the engagement of CD81 on NK cells by HCV E2 glycoprotein, and we have demonstrated it can be reverted by IFN-a treatment.
Point 4: Effector to Target ratios should be stipulated in the figure legends for all experiments or in the method section if kept constant.
Response 4: We have amended the manuscript in the methods section accordingly.
Point 5: Line 358: Please avoid using “(data not shown)”. If the data is not presented in the manuscript, the data should not be mentioned it in the results.
Response 5: we have amended the manuscript as requested
Point 6: If s-TRAIL is so induced in IFN-a treated NK cells (Figure 5A/ 5B), how do the authors explain that the treatment with neutralizing antibody did not yield any effect (Figure 4B/ 4C)?
Response 6: The direct role of s-TRAIL is to trigger apoptosis of target cells. In Fig. 4B/4C, we have quantified the amount of HCV-Ag released by HCV-Huh7.5 cell in the media and not HCV-Huh7.5 cell apoptosis. We are now showing an increase in HCV-Huh7.5 cell viability after treatment with IFNα-NKs in the presence of a neutralizing antibody against s-TRAIL (Supplementary Fig. 6), which indirectly indicates that s-TRAIL that is secreted by NKs can induce apoptosis of target cells.
Point 7: Figure 6A/B: Authors should validate the interaction between galectin-9 and HCV viral particle. Can the authors pellet galectin-9 when pelleting the virus by ultracentrifugation? Or using PEG? Be sure to include a negative control of Galectin-9 along in the media.
Response 7: We thank the reviewer for the thoughtful consideration. To address the question, we have used two different approaches: co-immunoprecipitation of human purified Galectin-9 and JFH1-HCV virions, and in silico prediction studies. Data are described in Fig. 6 and Supplementary Fig. 12, Supplementary Tables S1, S2, S3
Point 8: Figure 6C/D: Does co-culture of IFN-a treated NK cells with Huh7.5 cells trigger induction of MX1 and IFIT1 in comparison to co-culture with unstimulated or IL-2/IL-15 treated NK cells? It is important to validate that phenotype to confirm this is part of the response at play here.
Response 8: We thank the reviewer for the comment. We have now added western blots showing that target cells which are co-cultured in transwell with NK cells, upregulate MX1 only in the presence of IFNα-NKs and not in the presence of IL-2/IL-15-NKs (Fig. 7C and Supplementary Fig. 15).
Point 9: The authors often use t-tests in panels where they have more than 2 conditions. This is not the correct statistical test. When they compare more than two conditions, they should use 1- or 2-way ANOVA followed by a post-hoc pairwise test corrected for multiple comparisons such as Tukey or Bonferroni tests.
Response 9: We thank the reviewer for the suggestion. We have now included the suggested statistical analyses.
MINOR POINTS:
Point 1: Line 278: It is unclear if we are supposed to see the lipid droplets (LDs) in the histological work presented. Right now, the pictures are not of high quality enough to see intracellular details. Arrows indicating LDs to help the reader would be welcome.
Response 1: These images lost quality during conversion from powerpoint into PDF files. We have now used a different method to generate high quality images. Livers of uPA/SCID chimeric mice show areas of human hepatocytes intermingled with areas of mouse hepatocytes. As we have written in the text, human hepatocytes can be easy distinguished owing to a clear cytoplasm with glycogen storage and lipid droplets, as described in reference #19. We do not deem arrows indicating LDs necessary because we have already added letters indicating human (H) versus mouse (M) hepatocytes.
Point 2: Supplemental Figure 4A: Authors should keep these histograms format consistent with the other (Grey and color overlap).
Response 2: We have addressed this comment. Histograms have now been formatted
Point 3: Supplemental Figure 4B: Graphs appear to be pretty low quality; I can barely read the information on the axis.
Response 3: We have addressed this comment. These images lost quality during conversion from powerpoint into PDF files. We have now used a different method to generate high quality images.
Point 4: Supplemental Figure 4C: 3rd graph on the second line does not have the same y-axis as all the other graphs (Count instead of Count %). What is the meaning of count %?
Response 4: We have addressed this comment. All graphs now have the same format and label on the y axis (counts).
Reviewer 2 Report
This work studies the secreted molecules released from activated NK cells and their potential roles in suppressing HCV infection. Flow cytometry, cytokine profiling, and other commonly used analytic tools in immunology are used. The authors found that IFNα-NK cells (NK cells activated by IFNα) are superior in inhibiting HCV to IL2/IL15-NK. They also found that IFNα-NK cells express high levels of galectin-9 and IFN-γ. A few more exploratory experiments were performed to correlate these two molecules with the inhibition of HCV infection. The active role of galectin-9 in suppressing HCV infection is exciting and refreshing though it contradicts previous studies. The explanation of this conflict is acceptable. I appreciate the different opinions.
I have two minor suggestions.
1) There is no strong rationale for this study. The authors should clearly describe the knowledge gaps in the introduction to guide the audience.
2) The hypothesis of galectin-9 role in inhibiting HCV virus entry through direct interaction looks pretty immature. It could be misleading if there is no evidence.
Author Response
Point 1: There is no strong rationale for this study. The authors should clearly describe the knowledge gaps in the introduction to guide the audience.
Response 1: We thank the reviewer for his/her comment.
Improved understanding of the cellular and molecular mechanisms of cytokine-enhanced NK cell response to infection and cancer would lead to a resurgence of interest in the clinical use of cytokines that sustain and/or activate NK cell potential. Moreover, dissecting impaired NK cells function may help develop novel immunotherapeutic strategies.
We have now included the sentence in the introduction.
Point 2: The hypothesis of galectin-9 role in inhibiting HCV virus entry through direct interaction looks pretty immature. It could be misleading if there is no evidence.
Response 2: We thank the reviewer for his/her comment. To address this comment, first, we used in silico approaches to predict the binding of HCV to galectin-9; then, we performed a co-immunoprecipitation assay to confirm physical interaction. This new findings were included in the revised manuscript.
Round 2
Reviewer 1 Report
Authors sufficiently addressed my comments.